# Robust Tracking for Visual Complex Environments

## ABSTRACT

Achieving accurate tracking and robust tracking in visually complex scenes remains a challenging task. This requires to ensure that a robust appearance representation is captured while improving the generalization ability of the model to cope with challenges such as object deformation, illumination changes, scale changes, and motion blur. In this paper, we propose a robust tracking technique in complex tracking scenarios based on efficient convolution operator (ECO) tracker. It adopts two-fold ideas: a) extract deep features using the Conformer network after expanding the number of underlying channels, and b) adaptively adjust the fusion weight of shallow features and deep features according to the peak to sidelobe ratio and the joint score of adjacent frame trajectory smoothness. By doing so, the generalization ability of the tracking model and its adaptability in complex environments are improved, while making full use of the complementarity of deeper-layer and shallow-layer features. Experimental results show that the algorithm in this paper can effectively cope with different challenges of target tracking in complex environments, robustly tracking the target while maintaining high accuracy.

**Index Terms:** Computing methodologies—Computer graphics—Visual tracking

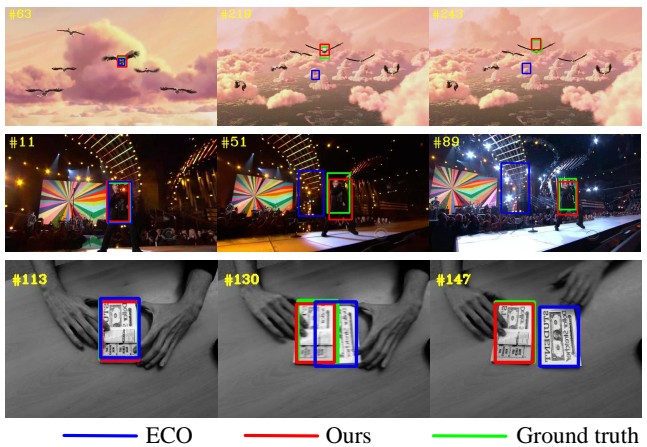

Figure 1: A comparison of our approach with baseline-ECO tracker. Observed from the visualization results, our tracker tracks objects more robustly than baseline-eco when encountering circumstances of complex environments.

## 1 INTRODUCTION

Visual object tracking refers to capturing real-time position, motion status and trajectory information of a target object by locating a specific target in a video sequence. Visual tracking is a fundamental and critical technology in the field of computer vision. It has extensive and important applications in intelligent security [47], traffic monitoring [21], unmanned driving [22] and other human-computer interaction issues. Tracking objects and backgrounds in complex environments (illumination variation, occlusion, fast motion, etc.) are often with unpredictable appearance changes, and the existence of these difficulties becomes a bottleneck that restricts the visual tracking algorithms towards practical applications.

The discriminative correlation filters (DCFs) [4] based target tracking algorithm has received wide attention in recent years due to its faster operation speed and higher tracking accuracy. To improve the robustness of discriminative correlation filter trackers in real complex scenarios, scholars have continued to explore and a large number of tracking algorithms have emerged. Based on the literature [4], the kernel function is introduced into the correlation filter [18], and it discards the single grayscale feature and uses the Histogram of Oriented Gradient (HOG) [7] feature. Compared with grayscale features, HOG features can be more robust target features for trackers, which can improve tracking accuracy. STAPE [1] uses two complementary feature factors, HOG features and color histogram, to learn the target and fuse the tracking results. To achieve a more discriminative image representation, DRT [34] introduces Colornames [15] features and suppresses tracking target background information. HOG and ColorNames features are the most commonly used target information features in correlation filtering algorithms in the past. This type of feature focuses on local, underlying, texture and contour information, but is susceptible to environmental change interference and target deformation.

In order to address the shortcomings of handcrafted features (Hog, ColorNames, etc.), many scholars introduce deep features [5] with richer semantic information into correlation filter trackers. For example, HCF [27] incorporated the hierarchical deep convolutional features for visual tracking. HDT [32] found that the features of different convolutional layers have different feature expressions, so it is proposed to use convolutional features to train correlation filters hierarchically. C-COT [13] investigated the problem of response map fusion due to different resolutions of depth features in different layers, which can effectively integrate multi-resolution depth feature maps. On the basis of C-COT, ECO [9] uses VGG-M [6] network to extract target features, and through factoring convolution operations and training set simplification, an efficient convolution operators tracking algorithm is proposed. However, ECO often leads to tracking drift in complex scenes, partly because the target features extracted by the VGG-M network are local features and cannot effectively represent the global features of targets in complex scenes [31]. Meanwhile ECO uses a fixed fusion weighting strategy to assign higher weights to deeper-layer features rich in semantic information and keep them dominant. This strategy lacks self-adaptability and cannot adjust the weights according to the environment to precisely locate the target.

In this paper, we proposed an algorithm to improve the general-

ization ability of the tracking model and the adaptability of complex scene tracking. First, we use the Conformer [31] network to extract target deeper-layer features. This network captures the deep features of the CNN local features combined with the Transformer [44] global features. This feature has richer global and local information, which is more suitable for visual tracking and can better capture the target characteristics after a drastic change of the target. And we improve the underlying structure of the network to enhance the ability to obtain texture features and contour features. Then, we propose a feature fusion strategy to cope with changes in the tracking environment. The strategy is to adaptively assign different weights to shallow-layer and deeper-layer features for response fusion by joint scoring of peak to sidelobe ratio (PSR) [4] between the adjacent frame trajectory smoothness.

The main contributions of our work can be summarized as follows:

- We improve the Conformer network to extract more robust target features.
- We propose a feature response map fusion strategy to cope with unpredictable changes in the appearance of tracked objects and backgrounds.
- To demonstrate the effectiveness of the tracking framework proposed in this paper, we conduct extensive experiments on three visual object tracking benchmark dataset UAV123 [29], OTB2015 [42] and OTB2013 [41].

## 2  RELATED WORK

In this section, feature extraction of DCF-based visual trackers is briefly explained. Then, we review the related work on adaptive feature fusion in tracking algorithm. For a thorough review, readers can refer to [23].

**DCF trackers.** In 2010, the Mosse [4] algorithm came out of the blue, and its excellent performance has led to extensive research on correlation filters in visual tracking. SRDCF [12] algorithm addresses boundary effects through spatial regularization for better performance in complex scenes. Benefiting from the closed solution of correlation filters, researchers try to jointly train filters and deep feature extraction networks. Classical works include CFNet [35] and DCFNet [40]. There are also more trackers that take full advantage of employing deep Convolutional Neural Networks (CNNs) that are pretrained on the ImageNet dataset [14]. An effective multi-cue analysis framework for deep feature tracking (MCCT) is proposed in [39], where correlation filters are trained separately using different layers of VGG-19 [33] features, and the best performing correlation filter is selected from the current frame for tracking. In order to reduce the redundancy of multi-channel features in the deep feature correlation filter and improve the discriminative ability of the selected features, the joint group feature selection and discriminative filter learning tracking algorithm [43] is proposed. In the ATOM [8] algorithm, the authors used a conjugate gradient strategy combined with a deep learning framework for fast optimization. Enables fast optimization of tracking models. The research team constrained the initialization filter to perform ridge regression loss in the work DiMP [2], so that the filter model has the ability of background discrimination in complex environments. Although end-to-end training can combine correlation filters and deep models, it requires a large dataset and time to refine the model.

**Adaptive feature fusion.** Limited by the training data of tracking, and the high dimensionality of the features, and the cost of model training, more trackers still use deep feature correlation filtering to achieve tracking. The DeepSRDCF [11] tracker uses VGG-M to extract deep features, and finds that deeper-layer features contain rich semantic information and are highly invariant, which is important for robustness in tracking. The shallow-layer features have rich texture information, which is crucial to the accuracy of tracking [3]. MCPFs [45] exploits the interdependencies among different features

to jointly derive relevant filters, and makes the learned filters complement and enhance each other for consistent responses. C-COT [13] uses continuous interpolation to effectively fuse deep feature maps of different resolutions, and achieves good results. In [43], GFS-DCF selectively fuses features at the spatial-channel-temporal level to achieve target-adaptive features. HCFTs [28] learn adaptive correlation filters on the outputs from each convolutional layer to encode the target appearance. The tracker infer the maximum response of each layer to locate targets in a coarse-to-fine manner. The above tracking algorithm uses CNNs to extract target features, which leads to missing global information about the target and cannot extract robust target features in complex tracking scenarios [31]. And use fixed weight fusion in the feature response map fusion stage. If there is semantic misleading, such as the occlusion of similar semantic objects, it will lead to the accumulation of errors in the training and updating process of the target model, which will affect the final decision of the model. Therefore, it is necessary to adaptively adjust the fusion weight of deeper-layer and shallow-layer features according to different tracking environments.

## 3  PROPOSED METHOD

In this section, we describe the proposed robust tracking for visually complex environments algorithm in detail. Our robust tracker component is basically devised complying with the following guidelines: (i) an excellent object feature extraction network Conformer to yield high-quality visual representation to retain sufficient target information for precise object boundary generation, (ii) improve the shallow-layer features of the Conformer to capture more texture and contour information of the target, and (iii) an efficient adaptive feature-response fusion strategy to handle appearance changes in complex continuous sequences. The framework of our method is depicted in Figure 2. As shown in Figure 2, the method in this paper mainly includes three steps: extracting features, obtaining response values, and adaptively fusing response values. Extract the shallow-layer features in the pre-trained Conformer model and the deeper-layer features after feature coupling, and use the implicit interpolation model to convolve the feature map with the correlation filtering to generate a response map. Get its highest response value. Then, using the adaptive deep feature fusion strategy proposed in this paper, the shallow-layer and deep-layer features response maps are adaptively fused to obtain the final response value and track the target.

### 3.1  Target Feature Extraction

CNNs [25] acts as a multilayer perceptron, and each convolutional layer can express different features of the input image. The shallow-layer contains more details, and the deeper-layer contains more semantic information. However, it is difficult for CNN to capture the global representation, and the background information cannot be well encoded into the target features in complex environments. Conformer [31] used convolutional operations and self-attention mechanisms to enhance representation learning. This network considered the feature mismatch between CNN and Transformer features designed Feature Coupling Unit (FCU), which fuses local features and global representations at different resolutions in an interactive manner.

In the Conformer network structure, the shallow layer (Conv_stem) retains a high spatial resolution, but the number of channels is small, resulting in the lack of texture and contour information. Considering the need to extract richer contour and texture information as target features in complex scene target tracking tasks. We make changes to the first layer of the Conformer network. The number of channels of the feature map is expanded, and the Local Response Normalization (LRN) layer is added after the convolution layer in order to increase the generalization and robustness of the model and suppress the neurons with smaller feedback.

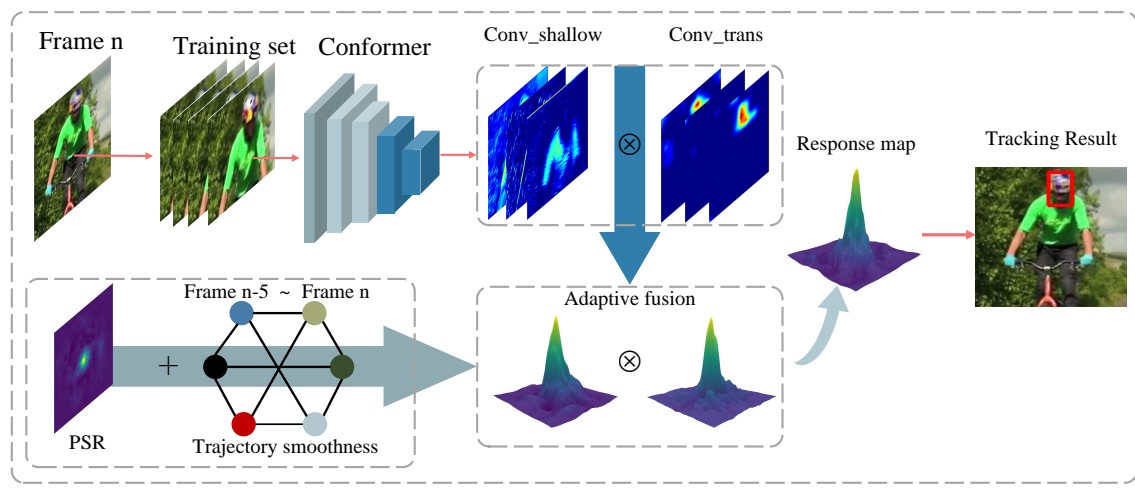

Figure 2: Pipeline of the proposed tracking framework.

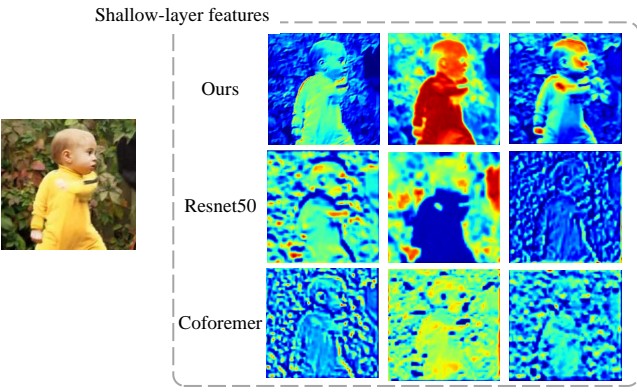

Figure 3: Visualization of Shallow feature map. Select the same channel for the control variable.

Let the current input image $x \in \mathbb{R}^{H \times W \times C}$. The convolution filter is $W$, and the input image is passed through the convolution filter $W$ to obtain 96 feature maps. The size of each convolution kernel in the convolution filter is 7*7, the number of channels is 3, and the stride is 2. The Relu function is selected as the activation function, and each feature map $x_i^2$ is obtained by Eq. (1).

$$x_i^2 = LRN\left(Relu\left(W_i \star x\right)\right) \tag{1}$$

where $\star$ represents three-dimensional convolution, and the LRN processing formula is shown in Eq. (2),

$$x_i^2 = \frac{x_i^1}{\left(k + \alpha \sum_{j=\max\left(0, i-\frac{n}{2}\right)}^{\min\left(N-1, i+\frac{n}{2}\right)} \left(x_i^1\right)^2\right)^{\beta}} \tag{2}$$

where $N$ is the total number of convolution kernels, and $x_i^1$ is the value after convolution operation and nonlinear excitation.

Figure 3 shows the visualization of shallow feature maps extracted by different networks. For the convenience of comparison, the feature maps of each layer are interpolated and converted to the same resolution. Compare Resnet50 [17] (Layer1), Conformer [31] (Conv_stem) and the modified shallow feature maps of the Conformer network in this paper. The improved shallow-layer features

have better spatial resolution than other networks, and are rich in texture and contour information. The details captured by it are of great help to the precise positioning of the target object of visual tracking. The deep-layer features of Conformer not only include local features extracted by CNN, but also contain global features extracted in Transformer. This means that the deeper-layer features in the Conformer have higher discriminative ability and more robust semantic features, which can deal with the interference caused by complex background changes and target deformation during the tracking process. The deep features extracted by Conformer can refer in [31].

### 3.2 Adaptive Feature Response Fusion

Unlike handcrafted features such as grayscale features and ColorNames features, depth features possess different layers. In the process of training the correlation filtering model, each layer of features has different effects on the tracking model. In the absence of significant appearance changes, the model relies mainly on the texture information and contour information of the shallow model. In the presence of challenging factors such as occlusion, deformation, and in-plane rotation, deeper-layer features sacrifice spatial resolution to increase high-order invariance to account for appearance changes compared to shallow-layer features. Deeper-layer features cannot be precisely located on the target. Not only is target pinpointing critical to tracking performance, but also has an impact on the update of the tracker model. Since the tracker itself annotates new frames, the introduction of incorrectly tracked targets can affect the model's judgment of tracked targets. Inaccurate predictions can lead to model drift and eventual tracking failure. Shallow-layer features are important for accurate target location, while deeper-layer features are critical for robustness.

For the extracted depth feature sample $x_j$, it contains $D$ dimensional feature. $N_d$ is denoted as the resolution per channel, $d \in 0, 1, 2, \ldots, D$ channels. Each layer of the input is represented as $x_j^d \in \mathbb{R}^{N_d}$, and the eigenvalues of each channel are denoted by $x_j^d[n] \, (n \subseteq [0, N_d - 1])$.

When deep features are introduced into object tracking, it represents a large training cost. For DCFs trackers, deep features bring benefits but also negative effects. The integration of high-dimensional feature maps leads to a sharp increase in the number of appearance model parameters, which often exceeds the dimension of the input image. In order to improve the running performance of the tracking algorithm, this paper adopts the ECO [9] tracking framework and uses factorized convolution operator to reduce the learning

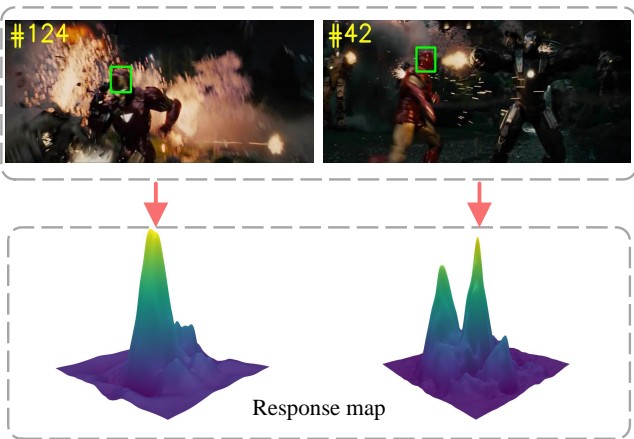

Figure 4: Peak response map comparison.

parameters in the correlation filtering model. The Conv_shallow layer (the first convolutional layer) and the Conv_trans_10 layer are extracted from the improved Conformer network as the feature map in the input correlation filter trainer, and the feature map size is 112*112 and 14*14 respectively. Using an implicit interpolation model, feature maps of different resolutions can be naturally integrated through a continuous convolution domain, and the period of continuous features is the same. Map feature maps in discrete space into a continuous spatial domain $t \in [0, T)$, where a constant T denotes the support size of feature map. For the feature with feature channel $d$, its interpolation operator $J_d$ is constructed as,

$$J_d \left\{ x^d \right\} (t) = \sum_{n=0}^{N_d - 1} x^d[n] b_d \left( t - \frac{T}{N_d} n \right) \quad (3)$$

where $n$ is index of spatial sample. $N_d$ denotes the number of spatial sample in $x_d$. The interpolation function $b_d$ is constructed based on cubic spline kernel.

The loss function is as follow:

$$E(f) = \sum_{k=1}^{T} \alpha_k \left\| \sum_{d=1}^{D} f^d * J_d \left\{ x^d \right\} - y_k \right\|^2 + \sum_{d=1}^{D} \left\| w f^d \right\|^2 \quad (4)$$

where * denotes convolution operator, $\alpha_k$ denotes the weight of sample in sample space, $f^d$ denotes the correlation filter and $w$ denotes regularization weights.

The convolutional responses of all feature channels are summed to obtain the final tracked target localization confidence $S_{Pf}$.

$$S_{Pf} \left\{ x \right\} = \sum_{d=1}^{D} f^d * J_d \left\{ x^d \right\} \quad (5)$$

Using Eq. (4), the maximum response value of the correlation filter can be found, and the Fourier inverse transform maximum response value is derived from the corresponding coordinate, which is the position of the target in the new image frame.

As mentioned earlier, deeper-layer and shallow-layer models have different characteristics in terms of accuracy and robustness. To ensure the accuracy and robustness of the tracking model, different fusion weights should be set for different tracking environments. The features of two different properties features are fused in an optimal way.

The peak response map can reflect the accuracy and robustness of target positioning. The accuracy is related to the sharpness of the predicted response around the target. The sharper the main peak, the

higher the accuracy; the robustness is related to the interval from the main peak to the interference peak. The larger the distance of the interference peak, the higher the robustness. As shown in Figure 4, if the tracking target is accurate, and there is no occlusion, deformation, etc., the peak response map shows a single peak corresponding to Figure 4 124-th frame, and the peak tip is relatively sharp. On the contrary, the tracker cannot accurately track the target, and the peak response graph shows multiple peaks, and the distance between the peaks is relatively close, corresponding to Figure 4 42-th frame.

The peak to sidelobe ratio (PSR) metric is a commonly used response map evaluation criterion in correlation filters, which represents the peak sharpness of the correlation filter response. The calculation equation is as follows,

$$P = \frac{(S_{\max} - m(S))}{\sigma(S)} \quad (6)$$

where $S_{max}$ represents the maximum response value in correlation filtering, and $m(S)$ and $\sigma(S)$ represent the mean and variance of the response values, respectively.

Calculate the PSR $P_d^i$ and $P_s^i$ of the shallow-layer and deeper-layer features of the i-th frame, respectively. During tracking, the tracking target may be lost, and the tracker may regard other objects in the search area as tracking targets, but the PSR may not change significantly at this time [37].

In [39], trajectory smoothness is used as an evaluation metric for tracking reliability. Inspired by it, this paper formulates a related frame smoothness score $O$ by measuring the target motion trajectory between the current frame and the previous 5 frames. The correlation frame smoothness scoring formula is given in the following equation,

$$O^j = \exp \left( - \left( \sum_{i=j-5}^{j} \left\| \frac{Q^i - Q^{i-1}}{\sqrt{2} \theta_i \times \eta^i} \right\| \right)^2 \right) \quad (7)$$

where $j$ is the current frame and $Q^i$ is the predicted bounding box center position information of the i-th frame. Since the correlation between the current frame and the previous frames is low, the correlation coefficient $\eta$ is set. $\eta_i$ is the mean value of the height and width of the predicted bounding box for the frame, and a higher value of $O$ indicates better tracking performance of the model.

In this paper, the PSR is combined with the trajectory smoothness score to obtain the fusion evaluation equation,

$$F^i = P^i \times O^i \quad (8)$$

$F_d^i$, $F_s^i$ denote the scores based on deeper-layer and shallow-layer features responses, respectively. The fusion score is obtained by a weighted combination of the two scores.

$$F_\beta^i = \max_{\beta_d, \beta_s} \beta_d F_d^i + \beta_s F_s^i \quad (9)$$

where $\beta = (\beta_d, \beta_s)$ are the fusion weights of the deeper-layer and shallow-layer features, respectively, both less than 1.

$$\text{minimize:} \quad L^i(\beta) = -F_\beta^i + \mu \left( \beta_d^2 + \beta_s^2 \right) \quad (10a)$$

$$\text{subject to:} \quad \beta_d + \beta_s = 1, \beta_d > 0, \beta_s > 0 \quad (10b)$$

where $\mu$ denotes the regularization term parameter, which penalizes large deviations of the weights. Convert it into a quadratic programming problem for calculation, and finally satisfy the $\beta$ with the smallest loss in Eq. (10) as the fusion weight. The fused correlation filter response as the following equation,

$$S_{\text{final}} = \beta_d S_d + \beta_s S_s \quad (11)$$

Table 1: OTB dataset description.

| Tracking environment | OTB2013 | OTB2015 |
|---|---|---|
| Illumination Variation (IV) | 22 | 38 |
| Scale Variation (SV) | 23 | 44 |
| Occlusion (OCC) | 38 | 64 |
| Deformation (DEF) | 11 | 14 |
| Motion Blur (MB) | 20 | 31 |
| Fast Motion (FM) | 8 | 9 |
| In-Plane Rotation (IPR) | 29 | 49 |
| Out-of-Plane Rotation (OPR) | 19 | 29 |
| Out-of-View (OV) | 25 | 39 |
| Background Clutters (BC) | 29 | 51 |
| Low Resolution (LR) | 32 | 63 |

Table 2: UAV123 dataset description.

| Tracking environment | number |
|---|---|
| Aspect Ratio Change (ARC) | 68 |
| Background Clutter (BC) | 21 |
| Illumination Variation (IV) | 31 |
| Out-of-View (OV) | 30 |
| Low Resolution (LR) | 48 |
| Full Occlusion (FOC) | 33 |
| Partial Occlusion (POC) | 73 |
| Fast Motion (FM) | 28 |
| Viewpoint Change (VC) | 60 |
| Similar Object (SOB) | 39 |
| Camera Motion (CM) | 70 |

## 4 EXPERIMENTS

In this section we introduces the experimental work, including experimental datasets, evaluation metrics, experimental environment and result analysis.

In the subsequent experiments, we aim at answering the following research questions (RQs):
a) **RQ.1**. Can the accuracy of the tracking algorithm in this paper be higher than the baseline ECO models and state-of-the-art trackers?
b) **RQ.2**. Can the proposed algorithm achieve robust tracking in complex environments?

### 4.1 Experimental Parameters and Datasets

**Experimental parameters.** The algorithm in this paper uses the python language and is implemented in the Pycharm integrated development environment. The experimental environment is python3.7.10, pytorch1.7.1, torchvision0.8.2, and Adamw [26] optimizer is used to train the modified Conformer model. The hyperparameters of LRN in Eq. (2) are set as $\alpha$=0.0005, $\beta$=0.75, $k$=2, $n$=5. The regularization term $\mu$=0.15 in Eq. (10a). The learning rate of the shallow feature correlation filter is 0.01, and the learning rate of the deep feature correlation filter is 0.0075. In Eq. (7), the correlation coefficient $\eta^i = 2$, $\eta^{i-1} = 4$, $\eta^{i-2} = 8$, $\eta^{i-3} = 16$, $\eta^{i-4} = 32$, $\eta^{i-5} = 64$.

**Datasets.** This paper uses the OTB-2013 [41], OTB-2015 [42] and UAV123 [29] datasets for test. (1) OTB-2013 contains 51 video sequences, this dataset is the first time to classify video sequences in different tracking environments. For illumination variation (IV), deformation (DEF), scale variation (SV), out of field of view (OV), background noise (BC), low resolution (LR) and so on. (2) OTB-2015 contains OTB-2013, which consists of 100 videos with 25% grayscale video sequences in the dataset. In [42], a large number of tracking algorithms are integrated and evaluated on the OTB-2015 dataset. (3) UAV123 consists of 123 low-altitude drone video sequences. Different from the OTB dataset, the shooting angle of the drone video changes greatly, the target is small, and the target exceeds the field of view for a long time, so compared with the OTB dataset, it is more difficult to track on this dataset. The classification of the dataset tracking environment is shown in Tables 1 and 2.

### 4.2 Comparisons on Track Benchmark

We use the One-Pass Evaluation (OPE) protocol as the evaluation protocol. Calculate the Distance Precision (DP), the Overlap Precision (OP) and the value of Area Under Curve (AUC) of different trackers as evaluation metrics. DP is the percentage of the number of frames whose Center Location Error (CLE) is greater than a certain Location Error Threshold (LET) to the total number of frames in the video sequence. The calculation equation of CLF is as follows,

$$C_{LF} = \sqrt{\left(x_p - x_g\right)^2 + \left(y_p - y_g\right)^2} \tag{12}$$

where $(x_g, y_g)$ denotes ground-truth central location, $(x_p, y_p)$ denotes prediction center location.

OP refers to the percentage of frames where the overlap rate $\phi$ of the tracking target frame $R^P$ and the ground-truth bounding box $R^G$ is greater than the over-lap threshold (OT) to the total number of frames.

$$\phi = \frac{\left|R^P \cap R^G\right|}{\left|R^P \cup R^G\right|} \tag{13}$$

where: $|\cdot|$ is the number of pixels in the region. In this paper, the LET is 20 and the OT is 0.5. LET is 20 and the OT is 0.5.

We compare our method to state-of-the-art tracker, including Ocean [46], MCCT [39], Dasima [48], UDT [38], ECO [9], C-COT [13], ATOM [8], MDNet [30], SRDCF [12], Siam_RPN++ [24] on OP metrics. The comparison results are shown in Table 3. The highest, second and third highest values are highlighted using red, blue and green respectively. The '-' in the table means that the tracker has not given official tracking data on this dataset. On the OTB2013 dataset, our tracker achieves the state-of-the-art performance of 95.20%, which is a 3.97% improvement over the base tracker ECO. On the OTB2015 dataset, the performance is also the best, reaching 88.69%, which is 2% higher than the base tracker ECO. Our tracker performs well even on the more complex UAV dataset, with 5.6% improvement over the base tracker ECO, respectively.

Next, we compare our approaches with state-of-the-art trackers on three datasets OTB2013, OTB2015, and UAV123. As shown in Figure 5, the tracking algorithm proposed in this paper outperforms all comparison algorithms on the OTB2013 and OTB2015 datasets. The DP and AUC of the algorithm in this paper are 1.7% and 2.2% higher than ECO tracker on the OTB2013 dataset; 1.7% and 1% higher than the ECO tracker on the OTB2015 dataset; and 6.1% and 3.5% higher than the ECO tracker on the UAV123 dataset, respectively. On the OTB2013 dataset, DP exceeds the second MDNet by 0.8%, and AUC is higher than the second ECO by 2.2%. On the OTB2015 dataset, DP exceeds the second C-COT by 1.6%, and AUC exceeds the second Siam_RPN++ by 0.4%. On the UAV dataset, our tracker greatly surpasses the classical correlation filter trackers KCF [18], DSST [10] and Struck [16], and is comparable to the DasiamRPN tracker that uses offline tracking. Compared with the offline trackers ATOM, Ocean, DasiamRPN and SiamRPN++, the tracker in this paper does not need to train a specific tracking network offline through a large number of tracking training sets.

### 4.3 Complex Environment Tracking Results

We compare different tracking algorithms on different tracking environments on the OTB2015 dataset. It is valuable to evaluate the performance of trackers. Evaluating different tracking environments can more intuitively reflect the robustness of a tracker to different challenging factors.

Table 3: Overlap precision comparison table.

| | UDT [38] | SRDCF [12] | ATOM [8] | DaSiamRPN [48] | Ocean [46] | C-COT [13] | MCCT [39] | MDnet [30] | Siam_RPN++ [24] | ECO [9] | Ours |
|---|---|---|---|---|---|---|---|---|---|---|---|
| OTB2013 | 75.78 | 78.37 | 83.74 | 87.74 | 87.88 | 88.50 | 89.09 | 91.13 | 89.76 | 91.23 | 95.20 |
| OTB2015 | 75.71 | 72.77 | 83.56 | 86.51 | 86.59 | 83.55 | 85.51 | 85.45 | 89.24 | 86.69 | 88.69 |
| UAV123 | - | 55.11 | 78.93 | 72.59 | - | 60.71 | - | - | 78.77 | 66.43 | 70.20 |

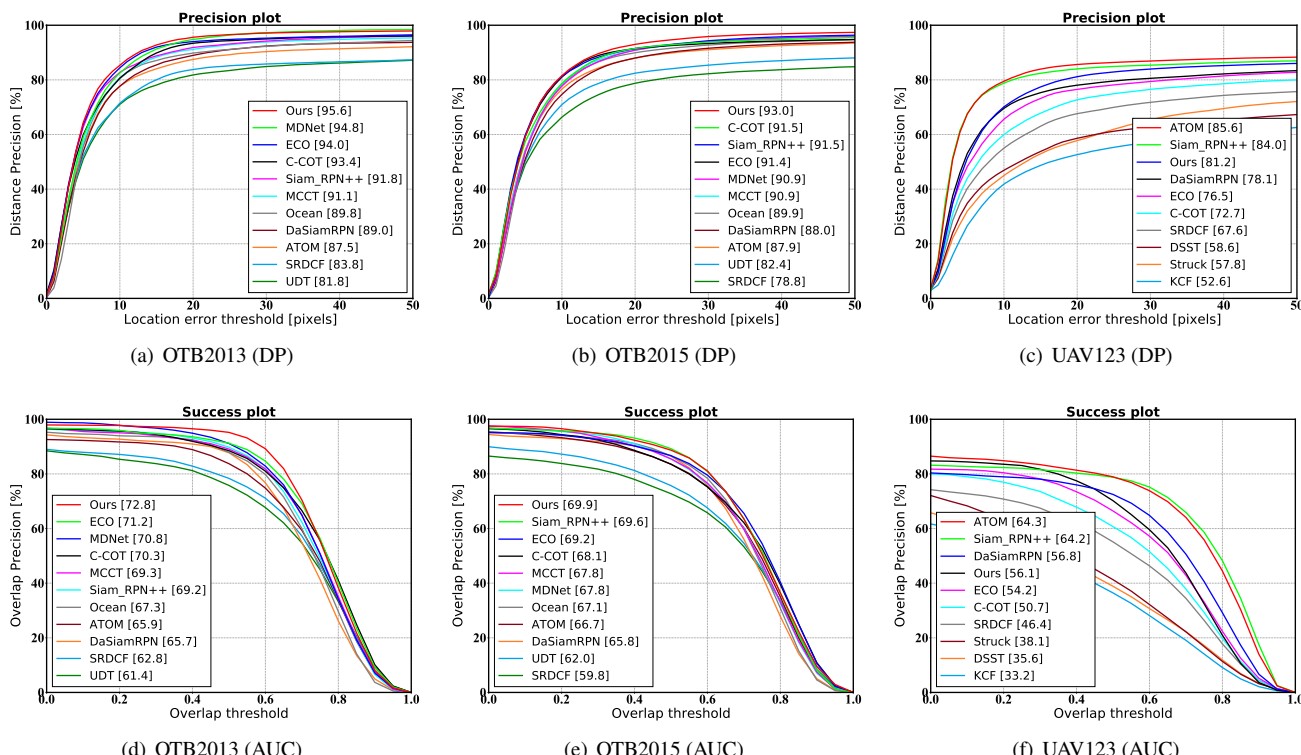

Figure 5: Success and precision plots of the state-of-the-art trackers on the three test set. The legend shows the AUC or DP score.

Table 4 shows the DP and AUC evaluation metrics values of the algorithms corresponding to Figure 5 under different tracking environments in the OTB2015 dataset. Our tracker performs well in 11 different tracking environments. The AUC scores of our tracker are in the top 3 in 10 of the 11 tracked environments, and it ranks first in almost all environments in the DP scores. Specially, our method achieves excellent performance on the sequences with attributes "Illumination Variation", "Occlusion", "Fast Motion", "Out-of-Plane Rotation" and "Out-of-View". For the above sequence of five tracking environments, our tracker ranks first in both evaluation metrics. It is proved that the proposed algorithm can track the target robustly and maintain high accuracy in complex environments. And our tracker has higher DP scores than the base tracker ECO except in the "Low Resolution" tracking environment. The reason can be attributed to the fact that this paper uses a more robust feature extraction method and improves the feature response fusion strategy. This enables the tracker to adjust the weights adaptively for different tracking environments and better adapt to complex environmental changes.

## 4.4 Ablation Study

In this section, we choose to verify the effectiveness of the enhancements to the tracker performance of the proposed components in this paper on the OTB2015 dataset. Extensive analysis of Conformer network and adaptive feature response fusion is performed.

**Conformer network.** In order to verify the effectiveness of the improved Conformer network structure proposed in this paper, it is compared with the currently popular networks with strong feature expression capabilities Resnet-50 [17], SENet [20], ECANet [36], MobileNetV3 [19] for comparison experiments. Based on the setting of ECO, the deeper-layer features respond and shallow-layer features respond are given fixed weights of 0.4 and 0.6, respectively. As shown in Table 5, simply increasing the depth or attention mechanism of the network cannot effectively improve the robustness and accuracy of the tracker. We observe that the Conformer network can achieve a good balance of robustness and accuracy. Benefit from the Conformer network that can extract local and global features, achieving excellent performance.

**Adaptive feature response fusion.** In order to verify the performance of the adaptive depth feature fusion proposed in this paper in target tracking applications, the DP and the AUC metrics are used to evaluate the OTB-2015 dataset. We divide into 4 cases for comparison: (1) only shallow features are used for tracking "Our_shallow", (2) only deep features are used for tracking "Our_deep", (3) adaptive fusion tracking "Our_fusion" and (4) fixed-weight fused tracking "Our_fixed". All the above algorithms use the Conformer network to extract features, and the parameter settings are consistent. The results for this analysis are shown in Figure 6. As shown in Figure 6, we can observe that without our adaptive feature response fusion strategy, simply use the conformer network to extract fea-

Table 4: 11 different tracking environments performance evaluation on the OTB2015.

| Algorithm | | IV | DEF | SV | OCC | MB | FM | IPR | OPR | OV | BC | LR |
|---|---|---|---|---|---|---|---|---|---|---|---|---|
| DSST [10] | AUC | 48.96 | 40.59 | 40.89 | 41.57 | 43.89 | 44.69 | 45.26 | 45.99 | 36.42 | 48.15 | 31.49 |
| | DP | 72.56 | 56.80 | 67.09 | 61.07 | 56.78 | 59.62 | 67.47 | 68.73 | 47.69 | 70.35 | 70.93 |
| KCF [18] | AUC | 47.92 | 43.62 | 39.55 | 44.61 | 45.84 | 47.75 | 45.49 | 47.13 | 39.30 | 49.76 | 29.00 |
| | DP | 71.87 | 61.66 | 63.96 | 63.20 | 59.78 | 63.84 | 67.56 | 69.70 | 49.51 | 71.23 | 66.51 |
| SRDCF [12] | AUC | 60.89 | 54.41 | 57.14 | 55.21 | 59.43 | 60.16 | 51.50 | 56.02 | 46.07 | 58.26 | 52.57 |
| | DP | 78.63 | 73.58 | 76.07 | 72.68 | 76.49 | 76.76 | 70.46 | 74.99 | 59.86 | 77.50 | 76.83 |
| MDnet [30] | AUC | 68.93 | 64.90 | 65.96 | 64.76 | 67.88 | 66.67 | 65.24 | 65.81 | 62.72 | 67.64 | 63.09 |
| | DP | 91.51 | 89.88 | 89.24 | 85.66 | 86.45 | 86.79 | 90.32 | 88.81 | 82.17 | 92.51 | 93.66 |
| C-COT [13] | AUC | 71.33 | 63.77 | 66.92 | 65.73 | 70.13 | 67.13 | 63.51 | 66.96 | 63.79 | 67.25 | 61.87 |
| | DP | 92.35 | 89.71 | 90.13 | 88.63 | 89.03 | 87.13 | 89.47 | 90.95 | 89.45 | 90.89 | 96.82 |
| ECO [9] | AUC | 71.03 | 63.19 | 68.08 | 66.26 | 71.27 | 68.36 | 65.08 | 66.95 | 67.21 | 65.59 | 67.63 |
| | DP | 92.17 | 86.51 | 90.44 | 88.08 | 90.02 | 88.60 | 90.04 | 89.07 | 90.13 | 86.72 | 100.00 |
| MCCT [39] | AUC | 68.50 | 63.47 | 65.04 | 64.63 | 66.89 | 65.55 | 64.12 | 66.48 | 64.28 | 70.01 | 66.57 |
| | DP | 88.62 | 88.08 | 88.94 | 85.98 | 85.66 | 87.58 | 89.80 | 89.54 | 86.26 | 92.53 | 100.00 |
| DasiamRPN [48] | AUC | 65.47 | 64.52 | 63.68 | 61.18 | 62.50 | 62.59 | 65.44 | 64.93 | 53.67 | 64.20 | 63.63 |
| | DP | 86.85 | 87.80 | 85.35 | 81.07 | 81.87 | 81.82 | 88.70 | 87.52 | 71.70 | 85.60 | 93.70 |
| UDT [38] | AUC | 59.68 | 57.81 | 60.34 | 58.42 | 62.50 | 58.44 | 52.80 | 58.37 | 59.21 | 61.07 | 50.04 |
| | DP | 76.30 | 78.70 | 81.10 | 77.71 | 78.94 | 74.66 | 72.43 | 79.42 | 77.60 | 81.29 | 76.30 |
| ATOM [8] | AUC | 66.10 | 63.24 | 67.16 | 63.52 | 65.04 | 64.56 | 64.70 | 62.85 | 59.31 | 60.78 | 70.31 |
| | DP | 86.98 | 85.97 | 87.91 | 83.05 | 83.26 | 82.65 | 86.58 | 84.20 | 79.69 | 79.13 | 99.25 |
| SiamRPN++ [24] | AUC | 71.33 | 66.28 | 69.44 | 66.68 | 70.42 | 68.99 | 69.91 | 68.42 | 64.82 | 69.12 | 69.91 |
| | DP | 92.48 | 89.48 | 91.75 | 87.80 | 90.74 | 89.30 | 93.34 | 91.43 | 84.90 | 90.32 | 99.25 |
| Ocean [46] | AUC | 68.86 | 65.16 | 67.55 | 63.68 | 68.39 | 66.49 | 69.15 | 66.19 | 64.08 | 62.11 | 66.44 |
| | DP | 91.44 | 91.16 | 90.90 | 86.02 | 90.39 | 88.60 | 92.67 | 89.25 | 87.09 | 82.87 | 96.67 |
| Ours | AUC | 73.08 | 65.50 | 67.16 | 67.16 | 69.95 | 69.52 | 68.32 | 68.67 | 67.27 | 69.72 | 64.82 |
| | DP | 94.08 | 90.43 | 91.10 | 89.04 | 91.51 | 92.10 | 94.94 | 92.39 | 93.01 | 91.34 | 99.03 |

Table 5: DP and AUC of different networks.

| Comparison of features | AUC (%) | DP (%) |
|---|---|---|
| Baseline | 69.1 | 91.4 |
| Baseline+ECAnet [36] | 53.1 | 72.3 |
| Baseline+SEnet [20] | 53.5 | 70.6 |
| Baseline+ResNet50 [17] | 68.3 | 90.2 |
| Baseline+MobileNetv3 [19] | 52.9 | 72.0 |
| Baseline+Conformer (not improved) [31] | 69.3 | 91.4 |
| Baseline+Conformer (ours) | 69.4 | 92.5 |

while being robust against distractor objects in the scene. An interesting direction for future work is to combine target segmentation techniques to address the situation where rapid scale changes in targets tend to cause tracking drift.

tures cannot achieve satisfactory results. The model tracked by the conformer network, which does not use adaptive feature response fusion, achieves an AUC score of 69.4%. The DP score reached 92.5%. The adaptive feature response fusion approach, which can exploit background information of different tracking environments, provides a substantial improvement, achieving an AUC and DP score of 69.9% and 93.0%, respectively. This highlights the importance of adaptive feature response fusion for model prediction.

## 5 CONCLUSION

In this paper, we propose a tracking algorithm suitable for application in complex scenes. Based on the Conformer network, deeper-layer features rich in global and local are extracted. To obtain more textured and contoured target features, we enlarge the number of shallow structure channels of the Conformer and add LRN normalization. Since shallow-layer features and deeper-layer features express different feature information respectively, and the target feature information is variable in complex environments, the use of fixed feature response fusion weights cannot cope with challenging tracking. To cope with feature changes in complex environments, we propose an adaptive feature response fusion method. Extensive ablation studies verify the effectiveness of the proposed techniques. Experiments on 11 different tracking environments and 3 tracking datasets show that our approach provided accurate target estimation

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
