# OpenReview forum: "Robust Tracking for Visual Complex Environments"
_graphicsinterface.org/Graphics_Interface/2022/Conference — Submitted to GI 2022_

### Official Review · Reviewer_9Zrn · 2022-04-08
**Combination of shallow and deeper features + fusion strategy based on peak-to-sidelobe ration and correlation smoothness**

**Rating:** 4
**Confidence:** 4

**Review:**

The authors present a robust tracking method based on the efficient convolution operator tracker. A Conformer network is used to extract features by combining the local features of the CNN with the global features of the Transformer. This, combined with a fusion strategy, leads to increased robustness when drastic changes occur to the target characteristics. Based on the DP and AUC metrics, the proposed method outperforms other methods marginally.

- It is stated that expanding the number of channels and introducing a Local Response Normalization layer after the convolutional layer increases generalization and robustness. I find the justification lacking and based on anecdotal evidence. It is well known that widening and deepening a network can improve its performance, and there is no discussion as to how the optimal number of channels is determined.

- It is difficult to reach conclusions based on visual comparisons of the activations at "shallow-layers" of different networks having different architecture. Depending on the architecture and the training schedule, networks may learn the same information at different depths. The only conclusion that one can draw from this comparison is that using the proposed method, the activations resemble higher-level information and are closer to the input image.

- k, \alpha, \beta in equation 2 are undefined. How are these derived?

- \theta_{i} in equation 7 is undefined

- How long does it take to solve for the weights for each bounding box, for each frame? Reporting on the time requirements will be beneficial to the readers.

- Equation 9 is very confusing. What is max? Do you mean argmax, and if so, why is there an argmax inside a function to be optimized? How are the constraints of equation 10 enforced by converting "it to a quadratic programming problem for calculation"? What type of bound-constraint optimization are you using?


Typos:
- Figure 3: "Conformerer"
- Section 4: "introduces"

---

### Official Review · Reviewer_6bZW · 2022-04-09
**Robust Tracking for Visual Complex Environments**

**Rating:** 4
**Confidence:** 5

**Review:**

The paper proposes a short term tracking approach based on the discriminative correlation filter (DCF) with features calculated by a transformer architecture. It therefore follows a current trend in improving short term tracking [1]. The authors state three main contributions including the adaption and improvement of the recent Conformer architecture by Peng et al. The Conformer combines the computation of low-level accurate detail features in a CNN with the power of transformers in finding global feature. The authors argue that tracking can benefit from these exact properties of the Conformer as precision is required for accurate tracking but global context for robustness of tracking. The modification of the Conformer architecture are limited to an increase in the number of channels of the feature map and the addition of Local Response Normalization after the convolution. The correlation filter response of the low level features and the global features are fused based on trajectory smoothness of the object being tracked. The fusion is accomplished by maximizing the peak-to-side-lobe-ratio (PSR), i.e., improving the uniqueness of the detection result for the current frame.

The proposed architecture is evaluated on OTB 2013 and 2015, as well as UAV123. The authors show that on OTB 2013 and 2015, their proposed tracker performs very well, often exceeding the performance of the comparator methods. Only brief results for UAV123 are given and appear less convincing. As the proposed tracker follows the general structure of ECO, ECO is used as a baseline for the ablation studies. In one study the expressiveness of the features are evaluated by integrating different CNNs into the baseline showing that the Conformer architecture outperforms various CNN architectures including ResNet50 for tracking. However, the differences between the baseline, the baseline with the original conformer architecture and the improved architecture are very small. Similarly, the improvement achieved by the fusion method is evaluated compared to using a shallow or a deep response map alone, as well as with fixed weighting between the maps. This study clearly shows the advantage of having a shallow and deep response map but the advantage of the fusion method over the fixed method is marginal.

Overall the paper describes a reasonable and fairly effective modification of an ECO based short term tracker. The improvement seems to be mainly a result of integrating the Conformer architecture and using a shallow and a deep feature response map. The novelty of the proposed approach is therefore somewhat limited. My main concern with the paper is however the fact that the paper ignores the VOT challenge with which the tracking community annually evaluates short-term tracking solutions. A careful review of this large-scale benchmarking effort, would have likely led to the use of the more challenging dataset from VOT and a revision of the list of comparator methods in Table 4. The run-time of the tracker is not discussed which is an important concern for short-term tracking as it is usual a building block in a larger system. It should be noted that the latest VOT challenge has demonstrated that the gap between real-time and offline trackers has considerably narrowed [1]. Another concern with the paper is the poor quality presentation with frequent expression and grammar errors which makes reading the paper unnecessarily difficult and introduces vagueness in the description.


[1] Matej Kristan, Jirı Matas, Aleš Leonardis, Michael Felsberg, Roman Pflugfelder, Joni-Kristian Kamarainen, Hyung Jin Chang, Martin Danelljan, Luka Čehovin Zajc, Alan Lukežič, Ondrej Drbohlav, Jani Kapyla, Gustav Hager, Song Yan, Jinyu Yang, Zhongqun Zhang, Gustavo Fernandez et. al.,
The Ninth Visual Object Tracking VOT2021 Challenge Results, VOT2021 challenge workshop, ICCV workshops, 2021

---

### Official Review · Reviewer_gpad · 2022-04-14
**Modifications on existing trackers with an unclear advantage**

**Rating:** 4
**Confidence:** 3

**Review:**

The paper addresses the problem of tracking objects in dynamic scenes with claims that the proposed modifications on existing architectures result in state-of-the-art performance.

First, I have several comments about the clarity. The title of the paper is very generic and does not effectively communicate the key ideas proposed. Furthermore, the text can make use of an extra proof reading pass, from small things like mistyping "Relu" to odd phrasing e.g. "Map feature maps", in order to increase clarity and flow.

Second, the main claim of performance increase does not seem to be matching with the results. In Table 3, the method seems to perform well on one out of three datasets compared to the state of the art. This indicate that the robustness claim is not backed by the presented evidence.

Lastly, the ablation study seems to indicate that the main reason behind the good performance is the use of the recent Conformer network. It has been well established that using new architecture designs often lead to better performance for any number of different visual tasks. I believe the paper needs to clearly show with experimentation how their proposal is adding some key novelty on top of the user of Conformer in the domain of tracking.

---

### Decision · Program_Chairs · 2022-04-17

Reject